# Histoplasmosis in Immunocompromised and Immunocompetent Patients in Guadeloupe

**DOI:** 10.3390/jof11060462

**Published:** 2025-06-18

**Authors:** Constance Lahuna, Tanguy Dequidt, Pierre Postel-Vinay, Sandrine Peugny, Marwan Haboub, Samuel Markowicz, Muriel Nicolas

**Affiliations:** 1Infectious Disease Department, Guadeloupe University Hospital, 97159 Pointe-à Pitre, France; tanguy.dequidt@chu-guadeloupe.fr (T.D.); sandrine.peugny@chu-guadeloupe.fr (S.P.); samuel.markowicz@chu-guadeloupe.fr (S.M.); 2Intensive Care Unit Department, Guadeloupe University Hospital, 97159 Pointe-à Pitre, France; pierre.postel.vinay@chu-guadeloupe.fr; 3Mycology Department, Guadeloupe University Hospital, 97159 Pointe-à Pitre, France; marwan.haboub@chu-guadeloupe.fr (M.H.); mu.nicolas@wanadoo.fr (M.N.)

**Keywords:** histoplasmosis, *Histoplasma capsulatum*, fungal infection, immunocompromised, immunocompetent patients

## Abstract

**Background:** *Histoplasma capsulatum* is an environmentally acquired dimorphic fungus. Infection results in histoplasmosis, a clinical syndrome often underdiagnosed and that may progress to life-threatening disseminated infection not only in immunocompromised individuals but also, following high-level exposure, in immunocompetent hosts. Epidemiological data from Caribbean regions, and particularly from Guadeloupe, remain limited. **Methods:** We performed a retrospective cohort study of all microbiologically confirmed histoplasmosis cases managed at the University Hospital of Guadeloupe between January 2014 and October 2024. Demographic, clinical, diagnostic, therapeutic, and outcome data were retrieved from medical records and analyzed using descriptive statistics. **Results:** Forty-two patients met the inclusion criteria, corresponding to an estimated annual incidence rate of 1 per 100,000 inhabitants. The median age was 52 years, and the male-to-female ratio was 4:1. An underlying immunocompromising condition was present in 85% of cases, most commonly HIV infection (48%). Common clinical features included weight loss (97%), fever (89%), and pulmonary manifestations (81%). The mean time to diagnosis from hospital admission was 3.5 ± 10.3 days. Direct microscopy was positive in 67% of cases, and culture was positive in 88% of cases. Intravenous liposomal amphotericin B constituted the initial therapy in 71% of patients. Overall, the in-hospital mortality was 29%, rising to 40% among HIV-positive individuals. The 30-day survival rate was 71%. **Conclusions:** Histoplasmosis in Guadeloupe is under-recognized and associated with appreciable morbidity and mortality in both immunocompromised and immunocompetent patients. The wider availability of rapid diagnostics and heightened clinical vigilance are essential to shorten diagnostic delays and improve outcomes in this Caribbean population.

## 1. Introduction

Histoplasmosis is a fungal infection caused by the dimorphic fungus *Histoplasma capsulatum*, which thrives in nutrient-rich soils, particularly those enriched by bat and bird droppings. Human infection typically occurs through the inhalation of aerosolized conidia [1].

*Histoplasma capsulatum var. capsulatum* is the only variant known to cause disease in humans throughout the Americas, including the Caribbean region [2]. While many infections are asymptomatic or mild, particularly in immunocompetent individuals, significant exposure, even in healthy hosts, can lead to acute pulmonary disease or disseminated forms. The risk of severe disease is significantly elevated in immunocompromised individuals, in whom delayed diagnosis and treatment can lead to high mortality [3].

Histoplasmosis is classified by the World Health Organization among neglected tropical diseases, under the category of “mycetoma, chromoblastomycosis, and other deep mycoses,” and is recognized as an AIDS-defining opportunistic infection [4]. Despite its significant morbidity and mortality, histoplasmosis is frequently underdiagnosed [5], particularly in resource-limited settings, and is often misdiagnosed as tuberculosis. This misclassification contributes to preventable mortality, particularly among immunocompromised populations [3], but may also impact immunocompetent individuals with high exposure risks. This highlights the urgent need for increased awareness and diagnostic capabilities.

Over the past two decades, the number of immunocompromised individuals has increased due to the broader use of immunosuppressive therapies, leading to a corresponding rise in opportunistic infections [6], including histoplasmosis [7].

In Guadeloupe, significant immigration from economically disadvantaged Caribbean nations has resulted in the presence of vulnerable populations with limited access to healthcare [8], placing them at a higher risk for histoplasmosis and other infectious diseases. Despite France’s publicly funded healthcare system, disparities in the service availability, particularly in rural or underserved areas, exacerbate these risks.

Despite the presence of the existing literature, contemporary studies on histoplasmosis in the Caribbean are limited and outdated [9], failing to capture the endemic nature of *Histoplasma capsulatum* among vulnerable populations, including HIV-positive individuals. While studies in other French territories, such as Martinique and French Guiana [10], have documented the presence of *Histoplasma capsulatum*, findings from these regions cannot be directly applied to the unique ecological context of Guadeloupe. This study aimed to address these gaps by investigating the incidence and clinical characteristics of histoplasmosis in both immunocompromised and immunocompetent patients in Guadeloupe, thereby contributing to a more accurate understanding of the disease’s epidemiology in this region.

## 2. Materials and Methods

### 2.1. Study Design and Setting

This retrospective, monocentric cohort study examined all the microbiologically confirmed histoplasmosis cases at the University Hospital of Guadeloupe (UHG) between 1 January 2014 and 1 October 2024.

### 2.2. Ethical Considerations

The study protocol received approval from the local ethics committee. Due to the retrospective and anonymous nature of the data collection, a waiver for informed consent was granted, in compliance with French legislation. Nonetheless, patients were informed about the ongoing study and given the option to decline participation. The study was registered under reference number A145_07/10/2024 at the Guadeloupe University Hospital.

### 2.3. Inclusion and Exclusion Criteria

Patients were included if they met the following criteria:Confirmed histoplasmosis: Diagnoses were based on the clinical presentation and microbiological confirmation (positive culture, PCR, or direct microscopic examination). In immunocompromised patients, the European Organization for Research and Treatment of Cancer/Mycoses Study Group (EORTC/MSG) criteria [11] were applied, incorporating host factors, clinical features, and mycological evidence. For immunocompetent individuals, diagnoses were established based on compatible clinical symptoms and definitive microbiological evidence, even in the absence of EORTC/MSG-defined host factors.Age ≥ 18 years.

The exclusion criteria included patients with unavailable medical records or a refusal to participate.

### 2.4. Data Collection

Data were gathered from positive *Histoplasma* cultures and direct microscopic examinations recorded in the mycology department’s database, as well as from PCR results obtained from the Toulouse PCR laboratory, a center part of the French National Reference Center for Invasive Mycoses and Antifungals, between 1 January 2014 and 31 October 2024. The following data were extracted from medical records:Demographics: Age, gender, and nationality.Socioeconomic factors and risk factors for exposure: Employment status, housing conditions, recent travel history, and substance abuse.Clinical presentation: Symptoms and severity criteria.Coinfections.Laboratory and imaging results.Patients’ HIV status and viral load, if applicable.Immunosuppressive conditions.Treatments received: Antifungals and steroids.Outcomes: Survival at 30 days and relapse within 3 months.

### 2.5. Diagnostic Modalities

Among the cohort, 37 patients were diagnosed by culture and 5 by PCR. While classical methods remained the primary diagnostic tools, PCR was increasingly used in recent years for its rapid turnaround time. However, no formal comparison of the methods’ sensitivity or accuracy was performed due to this study’s retrospective design.

### 2.6. Definitions

#### 2.6.1. Risk Factors for Exposure

Environmental exposure was evaluated based on occupational and recreational activities [12]. Occupational risks included soil exposure, especially through activities like construction, agriculture, gardening, and farming. Recreational activities included bird or bat contact or exposure to bat guano, such as through cave exploration or due to poor housing conditions and housing with barns or attics, considered to carry a heightened risk of exposure in the endemic setting of Guadeloupe [13].

#### 2.6.2. Social Vulnerability

Social vulnerability [14] was determined based on factors such as unstable employment, inadequate income, a lack of housing, a lack of a valid residency status, a lack of access to transportation, no healthcare coverage, and illiteracy. Substance abuse, defined as the current or past use of hard drugs (e.g., crack cocaine, heroin), chronic alcohol consumption, long-term tobacco use, or active cannabis use [15], was also considered.

#### 2.6.3. Severe Disease

Severe disease was defined according to the World Health Organization/Pan American Health Organization (WHO/PAHO) guidelines for histoplasmosis (Diagnosing and Managing Disseminated Histoplasmosis among People Living with HIV, 2020) [16]. For greater precision, we refined the severity criteria as follows:Pulmonary failure: The need for mechanical ventilation or high-flow oxygen therapy;Renal failure: Acute kidney injury stage III, according to the KDIGO criteria, and/or the need for acute kidney replacement therapy;Hemodynamic failure: A mean arterial pressure < 65 mmHg, lactate levels > 2 mmol/L, and/or the need for inotropic and/or vasopressor support (e.g., norepinephrine or dobutamine);Hematological failure: Disseminated intravascular coagulation (DIC), defined by a quick test (QT) result < 50% and platelet count < 50 G/L, and/or macrophage activation syndrome (MAS) requiring specific treatment (e.g., steroids or etoposide);Hepatic failure: A Factor V < 50% or symptomatic hypoglycemia requiring intravenous glucose;Neurological failure: A Glasgow Coma Scale score <8 or a persistent neurological deficit;

Underlying Conditions: Patients were classified into three categories [17]: (1) Patients with HIV; (2) individuals with other forms of immunodeficiency, including solid organ transplant (SOT) recipients, patients with active hematologic malignancies, and individuals receiving immunosuppressive therapies (e.g., TNF-α inhibitors, ibrutinib, T-cell-immunosuppressive drugs, B-cell-targeted therapies, anti-IL-6 treatments, or the long-term use of steroids > 5 mg/day for ≥3 months) [18]; (3) individuals without any identified immunodeficiency.

#### 2.6.4. Use of Steroids in Histoplasmosis Treatment

Steroid use was recorded and considered part of histoplasmosis treatment if administered at doses ≥40 mg/day of prednisone or its equivalent alongside antifungal therapy [19]. It was considered an aggravating factor when prescribed for a provisional diagnosis (e.g., sarcoidosis) while awaiting histoplasmosis confirmation.

#### 2.6.5. Clinical Outcomes and Relapse

A favorable outcome was defined as survival at 30 days following a histoplasmosis diagnosis. A relapse was defined as the recurrence of histoplasmosis symptoms within 3 months of initial recovery [20].

#### 2.6.6. Statistical Analysis

Statistical analyses were performed using Stata (version 12.1). Descriptive statistics were presented as the mean ± the standard deviation (SD) for continuous variables and as frequencies and percentages for categorical variables. The Wilcoxon rank sum test was used for continuous variables with non-normal distributions, while categorical variables were analyzed using Fisher’s exact test. A *p*-value < 0.05 was considered statistically significant.

## 3. Results

### 3.1. General Data

During the study period (January 2014–October 2024), the institutional microbiology registry recorded 43 histoplasmosis cases; one was subsequently excluded after diagnostic revision, leaving 42 analyzable patients. The registry revealed a global incidence rate of 1/100.000 inhabitants per year. The mean age of the patients was 52 ± 15 years, with a male-to-female ratio of 4:1. The majority of the patients were admitted to the infectious disease unit (70%, 29/42), while the remainder were hospitalized in pulmonology (14%, 6/42), nephrology (10%, 4/42), or other specialized units (14%, 6/42). A total of 74% (31/42) of the patients were French nationals, while 26% (11/42) originated from other Caribbean nations, including Dominica, Haiti, and Jamaica. Socioeconomic risk factors were identified in 40% (13/33) of the patients. Of the 33 patients with documented substance exposure, 50% (17/33) had a history of substance abuse, including the use of tobacco (36%, 12/33), alcohol (30%, 10/33), crack/cocaine (12%, 4/33), and cannabis (9%, 3/33).

### 3.2. Immunosuppression

An immunocompromised condition was identified in 76% (32/42) of the patients. Of these, 62.5% (20/42) were HIV-positive with a mean CD4 count of 49 ±  45 cells mm^−3^. Other causes of immunosuppression were identified in 37.5% (12/32) of the patients, including a SOT in 19% (6/32) of cases and autoimmune diseases or hematological malignancies in 9% (3/32) of cases. In 9% of cases (3/32), the diagnosis of histoplasmosis led to the identification of idiopathic CD4+ T lymphopenia. No monogenic immune defects were confirmed.

Six patients (14%) presented with primary histoplasmosis without an identifiable immunodeficiency, four of whom belonged to a familial cluster. Additionally, 17% (7/42) of the patients had comorbid conditions, including uncontrolled diabetes, advanced chronic kidney disease, sickle cell disease, active cancer, liver cirrhosis, and severe pulmonary emphysema, which may have contributed to the infection’s severity.

The remaining 21% of the patients (9/42) had no documented immunosuppression or other relevant comorbidities.

### 3.3. Environmental Exposure

Recent exposure to *Histoplasma capsulatum* was documented in 80% of patients with available data (16/20). The reported sources comprised occupational activities (*n* = 4), recreational activities (*n* = 8), and contact with animal droppings in the home environment (*n* = 4).

### 3.4. Clinical Symptoms

At presentation, the most frequent symptoms were weight loss (97%, 41/42), fever (89%, 37/42), and pulmonary manifestations (81%, 34/42) (Figure 1a). Extra-pulmonary disease (defined as confirmed *Histoplasma* at at least one non-pulmonary site or compatible lesions identified using imaging/biopsy) was documented in 50% of cases (21/42), most commonly with lymph node, hepatic, splenic, or bone marrow involvement. No case fulfilled the strict criteria for proven central nervous system (CNS) histoplasmosis as head imaging abnormalities in ten patients could not be attributed definitively to *Histoplasma*.

The mean time to diagnosis from hospital admission was 3.5 ± 10.3 days, and all but one patient was diagnosed during their initial hospitalization.

Overall, 60% of the patients (25/42) required admission to the intensive care unit (ICU). Among these, one patient did not meet the predefined severity criteria. Nine of the ICU patients (36%) were admitted directly to the ICU upon initial presentation. Respiratory failure was the leading indication for critical care (70%, 18/25), followed by neurological (48%, 12/25) and hemodynamic failures (43%, 11/25) (Figure 1b). Multisystem organ failure accounted for 7% of deaths (3/42), and in all three cases, histoplasmosis was diagnosed post mortem.

Coinfection was documented in 53% of the patients (20/38) at the time of histoplasmosis diagnosis. Within this subgroup, viral reactivation occurred in 37% of cases (7/20), bacterial or mycobacterial infection in 37% of cases (7/20), non-HIV viral disease in 30% of cases (6/20), fungal infection in 25% of cases (5/20), and parasitic infection in 10% of cases (2/20) (Table A1).

### 3.5. Biological Parameters

Biological parameters, summarized in Table 1, were compared across three subgroups: HIV-positive patients, immunosuppressed HIV-negative patients, and patients presumed immunocompetent.

HIV-positive patients and immunosuppressed HIV-negative patients showed significant differences in their aspartate aminotransferase (AST) (eight vs. two times the upper value, *p* = 0.03), bilirubin (18 vs. 22 μmol/L, *p* = 0.04), lactate dehydrogenase (LDH) (seven vs. three times the upper value, *p* = 0.02), and fibrinogen levels (3 vs. 5 g/L, *p* = 0.02).

A comparison between HIV-positive patients and those without immunosuppression revealed significant differences in their hemoglobin levels (10 vs. 12 g/dL, *p* = 0.02), platelet count (150 vs. 279 G/L, *p* = 0.01), lymphocyte count (0.7 vs. 1.3 G/L, *p* = 0.02), activated partial thromboplastin time (aPTT) (1.3 vs. 0.95, *p* = 0.002), AST (8 vs. 1.6 times the upper value, *p* = 0.02), LDH (7 vs. 2 times the upper value, *p* = 0.002), ferritin (23,123 vs. 1322 ng/mL, *p* = 0.003), and fibrinogen levels (3 vs. 5 g/L, *p* = 0.02).

Immunosuppressed HIV-negative patients and patients presumed immunocompetent displayed differences in their platelet count (135 vs. 279 G/L, *p* = 0.009), lymphocyte count (0.7 vs. 1.3 G/L, *p* = 0.03), bilirubin levels (22 vs. 15
μmol/L,
*p* = 0.02), and ferritin levels (20,168 vs. 1322 ng/mL, *p* = 0.002).

### 3.6. Imaging Data

All the patients underwent imaging during hospitalization, including chest radiographs or thoracic CT scans. The main imaging findings are summarized in Figure 2.

### 3.7. Microbial Results

Microbiological findings from all 42 patients showed that 67% (28/42) of the patients had positive samples based on direct examination, 88% (37/42) had positive samples in culture, and 48% (10/21) had positive samples when using PCR. All the microbial results related to *Histoplasma* are shown in Table 2.

### 3.8. Initial Therapeutic Strategies

Among the 38 patients with complete treatment data, 8% (3/38) died before antifungal therapy could be initiated. Intravenous liposomal amphotericin B was administered to 79% of the patients (30/38), whereas itraconazole (oral or intravenous) was used in 13% of the patients (5/38). Among those receiving amphotericin B, 80% (24/30) received 3 mg kg⁻^1^ day⁻^1^ and 20% (6/30) received 4–5 mg kg⁻^1^ day⁻^1^. The itraconazole doses ranged from 200 mg day⁻^1^ to 600 mg day⁻^1^. Adjunctive corticosteroids were prescribed for 38% (13/34) of the evaluable patients: hydrocortisone hemisuccinate in 23% (3/13), methylprednisolone in 46% (6/13), prednisolone in 15% (2/13), and an unspecified preparation in 15% (2/13) of these patients. Apart from in three clearly indicated cases with a solid organ transplantation, pneumocystis pneumonia, and a suspected adrenal insufficiency, the corticosteroid use largely reflected a local practice of co-prescribing steroids with antifungal therapy at diagnosis.

Pre-diagnostic corticosteroid exposure (0.5–1 mg kg⁻^1^ prednisone equivalents) was documented in four patients and was considered to have aggravated the disease severity. Two of these individuals had been treated for presumed sarcoidosis, leading to clinical deterioration and the eventual diagnosis of histoplasmosis.

### 3.9. Maintenance Therapy and Follow-Up

Among the survivors, 64% of the patients (25/39) received maintenance therapy, predominantly itraconazole (96%, 24/25). Two were switched to isavuconazole due to drug interactions or poor itraconazole tolerance. The duration of the therapy ranged from 90 to 365 days, based on the patient’s immunocompromised status. Five patients were treated for 90 days owing to the absence of immunosuppression, and three received secondary prophylaxis with itraconazole because their CD4⁺ count was below 200 cells mm⁻^3^.

The in-hospital mortality rate was 29% (12/42). Compared with the survivors, the non-survivors presented with significantly greater severity at admission (*p* = 0.002), were more likely to require ICU admission (*p* = 0.003), and differed in their ferritin (*p* = 0.04), fibrinogen (*p* = 0.02), and albumin levels (*p* = 0.004). No other parameters differed significantly between the groups.

At 30 days, 71% of the patients (30/42) were alive, although one subsequently died two months later in hospital. One patient developed immune reconstitution inflammatory syndrome (IRIS). At a one-year follow-up, 95% of the patients (19/20) with available data demonstrated favorable clinical evolution without a relapse.

### 3.10. Comparison of Most Relevant Results Across Three Immunosuppression Subgroups

Social vulnerability was significantly more prevalent among HIV-positive patients than in the other two subgroups (*p* = 0.007–0.009; Table 3).

Both immunosuppressed cohorts (HIV-positive and HIV-negative) showed higher proportions of positive blood cultures than the group presumed immunocompetent (*p* = 0.03–0.04) and exhibited greater rates of positive direct microscopic examinations. Liposomal amphotericin B was more frequently employed as a first-line therapy in these two groups (*p* = 0.03–0.05). Although the mortality in HIV-positive patients was roughly twice that observed in the other groups (40% vs. 17% and 20%), the difference was not statistically significant (*p* = 0.52). No additional variables differed significantly among the three subgroups.

## 4. Discussion

The crude annual incidence rate of histoplasmosis in our cohort was 1/100,000 population per year. Although these data were derived solely from samples processed at the University Hospital of Guadeloupe (UHG) and may therefore not be fully generalizable, the rate was ∼100-fold higher than that reported for mainland France (0.01/100,000) [21] and 7-fold lower than the figure for French Guiana [22]. The increase relative to historical data from Martinique, which reported 1.25 cases per year (0.25/100,000) [9], probably reflects an improved diagnostic capacity and the wider use of immunosuppressive therapies.

The mean age was 52 years, with a male-to-female ratio of 4:1—lower than the 9:1 ratio described in Martinique [9]. While a sex-related susceptibility to some endemic mycoses (e.g., estrogen-mediated resistance to paracoccidioidomycosis [23]) has been proposed, the male predominance in our study more likely mirrors the higher regional prevalence of HIV infection among men, a principal risk factor for disseminated disease.

Immunocompromised patients accounted for 78% of the symptomatic cases, predominantly patients with HIV, echoing observations from South America [24]. In this subgroup, 80% exhibited marked social vulnerability, compared with 8–10% of the other immunosuppressed or immunocompetent patients. Nevertheless, 24% of cases occurred in ostensibly immunocompetent individuals, 40% of whom required ICU admission, presumably owing to substantial fungal exposure. Idiopathic CD4⁺ lymphopenia, a rare condition with an estimated prevalence of 0.02–0.05% [25], was confirmed in two patients and suspected in a third, suggesting a higher-than-expected frequency. Because an idiopathic CD4⁺ lymphopenia diagnosis is often retrospective, a comprehensive immunological work-up and longitudinal follow-up are essential. As in HIV infection, prolonged antifungal therapy lasting up to one year may be warranted when the CD4 counts remain < 150 cells mm⁻^3^.

The clinical manifestations were non-specific, consistent with prior reports [26,27], and no pathognomonic feature emerged. Cutaneous or mucosal lesions, classically described in immunocompetent hosts [28], were documented in only eight immunocompromised patients; most were not biopsied, precluding the confirmation of *Histoplasma* involvement. No cases of adrenal insufficiency were observed, contrary to some South and Central American series [27].

Head imaging abnormalities were noted only descriptively and could not be linked definitively to histoplasmosis. Cerebrospinal fluid cultures, incubated for 30 days, were negative in all ten patients who underwent lumbar puncture; thus, CNS involvement remained unproven in our cohort. In HIV-positive individuals, alternative opportunistic infections, such as toxoplasmosis, may partially explain these radiological findings.

Molecular studies have identified distinct genotypes within *H. capsulatum var. capsulatum* [2,29]. The genotype-specific virulence may underlie geographic differences in clinical presentations and deserves further investigation.

The laboratory findings revealed a pronounced systemic inflammatory response with cytopenia, elevated transaminases, LDH, triglycerides, and ferritin, and hyponatremia, features suggestive of MAS. MAS was confirmed in at least eight patients and was most prominent in the HIV-positive subgroup, in line with previous work linking histoplasmosis-associated MAS to increased mortality [30]. Notably, the fibrinogen levels remained within normal ranges despite the inflammatory state. No MAS-related abnormalities were observed in patients presumed immunocompetent. These results correspond with previous research indicating MAS to be a severe complication of histoplasmosis, especially in HIV-positive patients, with a high mortality rate of 14% overall and 10% in histoplasmosis-related cases [30].

The culture positivity was high (88%), as was the positivity based on direct examination (67%). Nevertheless, 16.5% of bronchoalveolar lavage, blood, and bone marrow samples were contaminated with *Candida* or *Aspergillus*, underscoring the risk of false-negative cultures in tropical settings rich in environmental fungi [31]. PCR yielded rapid and highly sensitive results [32], whereas culture required at least 14 days [33]. Establishing local PCR capabilities would shorten the time to diagnosis, which is particularly important given that 7% of patients died before confirmation. Previous studies [34] have shown that serological markers like galactomannan offer limited utility, often lacking specificity and yielding inconsistent results. For example, the serum galactomannan was positive in only 48% of cases, and the BAL galactomannan was positive in 64%. These findings support current recommendations against the routine use of these markers for diagnosing histoplasmosis. *Histoplasma* antigen detection was unavailable in France during the study period.

Most patients received liposomal amphotericin B, in line with the WHO 2020 guidelines for severe disease [16], which reserve itraconazole for milder cases [16]. One patient who fulfilled the severity criteria received itraconazole alone and succumbed, emphasizing the importance of an appropriate initial therapy. The application of the prognostic score proposed by Francoise et al. [35] could help identify patients who might safely receive itraconazole, sparing others the nephrotoxic risks of amphotericin B [36].

Adjunctive corticosteroids were given to 38% of patients, yet no association with the in-hospital mortality was detected. The 2011 American Thoracic Society (ATS) guidance [19] recommends corticosteroid therapy only for narrowly defined scenarios, immunocompetent patients with severe hypoxemia and diffuse pulmonary infiltrates or with massive granulomatous mediastinitis, so its broader use remains controversial. In immunocompromised hosts, the risk–benefit balance is further complicated by IRIS: the ATS cites only four confirmed IRIS cases in which steroids were used therapeutically and not prophylactically [37], underscoring the limited evidence base. Epidemiological data likewise suggest that IRIS is uncommon in histoplasmosis: Melzani et al. reported 0.74 cases per 1000 HIV-positive person-years [38], and only one episode occurred in our cohort. Adrenal insufficiency was also rare, appearing less often than the 7% prevalence noted in the ATS document [19]. Coinfection was nevertheless frequent, especially in HIV-positive patients with profound CD4 cell depletion, who showed a mean of two concomitant infections and an overall coinfection rate of 28%. Although corticosteroid use was not linked to excess mortality in our series, steroids can potentiate fungal proliferation and aggravate coinfections [39]. Given the low incidence of IRIS, the limited evidence for benefits, and the high burden of concomitant infections, we suggest restricting corticosteroid therapy to well-validated indications such as documented adrenal insufficiency, refractory septic shock, or established IRIS.

HIV-positive patients showed a 40% case fatality rate, approximately twice that of the other subgroups, although this difference was not statistically significant. This figure mirrors observations from South America [40] and implies that determinants beyond immunosuppression, namely social vulnerability, delayed access to care, and a heavier burden of concomitant coinfections (a mean of 2.0 versus 0.2 and 0.3 in the other groups), may underlie the higher mortality. Although the crude mortality appeared higher in patients with concomitant infections (35% vs. 28%), the difference was not significant (*p* = 0.66), very likely reflecting limited statistical power. Consistent with these findings, 69% of HIV-positive individuals exhibited social vulnerabilities, circumstances associated with therapeutic non-adherence [15], lower CD4 counts, and an elevated risk of opportunistic infections. We believe that improving access to social resources for HIV-positive patients, including shelters with medical support and therapeutic housing for those with chronic health issues, would be highly beneficial.

This study has several limitations, primarily related to its retrospective design and relatively small sample size, which limited the consistency of the clinical information collected. Although we were able to retrieve a substantial amount of information from medical records, some degree of missing or inconsistent data was unavoidable. To minimize interpretation bias, we systematically collected and reported all the available clinical, biological, imaging, and microbiological findings, regardless of whether a direct causal relationship with histoplasmosis was suspected.

The high prevalence of comorbidities and coinfections, particularly among immunocompromised individuals, made it challenging to assess the specific impact of histoplasmosis on the patient outcomes, especially the mortality. While coinfections may have contributed to the observed mortality rate, we deliberately refrained from attributing deaths directly to histoplasmosis in the absence of clear evidence.

Additionally, this study likely underestimated the true incidence of histoplasmosis, as inclusion was restricted to microbiologically or molecularly confirmed cases (via culture, histopathology, or PCR). The absence of antigen detection and the limited use of serological testing likely resulted in the exclusion of patients with clinically probable disease but without confirmatory laboratory evidence. This highlights the broader diagnostic limitations in resource-constrained settings and underscores the need for improved access to sensitive, rapid diagnostic tools.

## 5. Conclusions

Histoplasmosis remains underdiagnosed, and its incidence is expected to rise as the use of immunosuppressive therapies expands. In our Caribbean cohort the infection rate was high, and more than half of the cases were classified as severe, even among immunocompetent patients, while HIV-infected individuals accounted for the most fatalities. Conventional culture- and microscopy-based diagnostics remain reliable but are inherently slow. Incorporating rapid assays such as PCR or urine antigen detection, although these are not necessarily more sensitive, could shorten the time to diagnosis, enable earlier treatment, and ultimately lower the mortality.

Finally, the therapeutic role of corticosteroids warrants re-evaluation, especially in the highly vulnerable HIV-positive subgroup.

## Figures and Tables

**Figure 1 jof-11-00462-f001:**
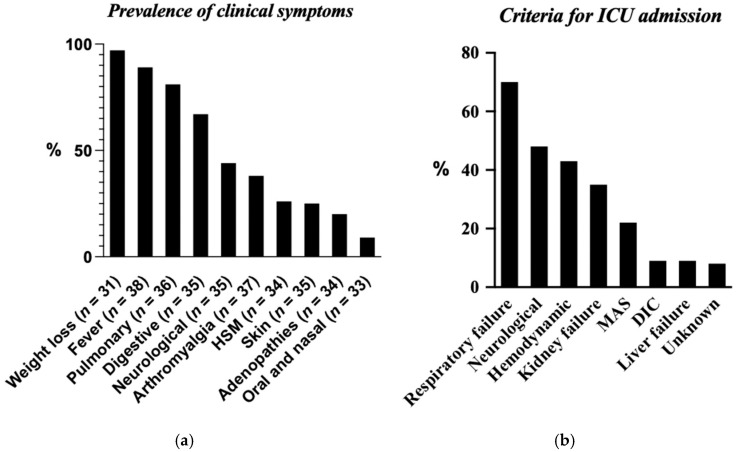
(**a**) Prevalence of clinical symptoms. (**b**) Criteria for ICU admission. HSM: Hepato-splenomegaly; MAS: macrophage activation syndrome; DIC: disseminated intravascular coagulation. N: number of patients for whom data were available.

**Figure 2 jof-11-00462-f002:**
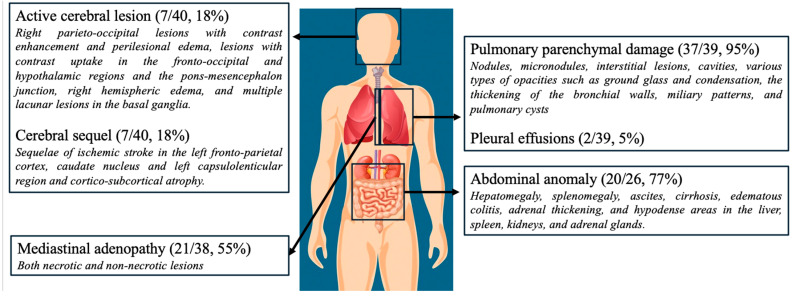
Imaging findings. CNS involvement included cerebral sequelae (7 patients) and active lesions (7 patients), corresponding to a total of 10 patients, with some exhibiting both types of lesions.

**Table 1 jof-11-00462-t001:** Biological parameters in overall cohort and according to immune status.

Biological Parameters	Overall Cohort (*N* = 42)	HIV-Positive (*N* = 20)	HIV-Negative, with Other Cause of Immunosuppression (*N* = 12)	HIV-Negative, Presumed Immunocompetent (*N* = 10)
Hemoglobin (g/dL)	10.5 ± 2	10 ± 2	10.7 ± 2	12 ± 2
Platelets (G/L)	181 ± 120	160 ± 105	135 ± 80	279 ± 140
WBCs (G/L)	5 ± 2.5	4 ± 3	5 ± 2	6 ± 2
Lymphocytes (G/L)	0.8 ± 0.6	0.7 ± 0.5	0.7 ± 0.5	1.3 ± 0.7
Neutrophils (G/L)	3.6 ± 2 (*n* = 41)	3.5 ± 2	3.4 ± 2 (*n* = 11)	4 ± 2
QT (%)	76 ± 15 (*n* = 41)	73 ± 18	77 ± 12 (*n* = 11)	79 ± 12
aPTT	1.2 ± 0.3 (*n* = 41)	1.3 ± 0.3	1.2 ± 0.3 (*n* = 11)	0.95 ± 0.1
AST	5 ± 11 (*n* = 41)	8 ± 16	2 ± 2.4 (*n* = 11)	1.6 ± 0.8
ALT	2 ± 3 (*n* = 41)	2 ± 4	1.7 ± 1 (*n* = 11)	2 ± 1.7
GGT	3 ± 4 (*n* = 41)	4 ± 5	4 ± 3 (*n* = 11)	2.5 ± 4
ALP	4 ± 12 (*n* = 41)	6 ± 17	2 ± 1.4 (*n* = 11)	1.6 ± 1.5
Bilirubinemia (μmol/L)	18 ± 16.5 (*n* = 41)	18 ± 20	22 ± 12 (*n* = 11)	15 ± 14
CRP (mg/L)	126 ± 100	126 ± 118	133 ± 57	118 ± 107
eGFR (ml/min)	69 ± 32 (*n* = 41)	73 ± 31 (*n* = 19)	52 ± 32	84 ± 28
CPK	1 ± 0.5 (*n* = 34)	1.3 ± 0.7 (*n* = 15)	1.1 ± 0.3 (*n* = 11)	1 ± 0 (*n* = 8)
Albuminemia (g/L)	28 ± 8 (*n* = 38)	26 ± 7 (*n* = 18)	29 ± 7	32 ± 10 (*n* = 8)
LDH	5 ± 6 (*n* = 37)	7 ± 7 (*n* = 19)	3 ± 1.4 (*n* = 10)	2 ± 0.7 (*n* = 8)
Triglycerides (mmol/L)	3 ± 4 (*n* = 31)	2.2 ± 1 (*n* = 18)	5 ± 7.6 (*n* = 9)	1.7 ± 1.3 (*n* = 4)
Natremia (mmol/L)	131 ± 5	130 ± 4	131 ± 7	133 ± 4
Ferritinemia (ng/mL)	18,586 ± 17,120 (*n* = 36)	23,123 ± 17,008 (*n* = 19)	20,168 ± 16,852 (*n* = 11)	1322 ± 1061 (*n* = 6)
Fibrinogen (g/L)	4 ± 2 (*n* = 32)	3 ± 1.7 (*n* = 16)	5 ± 1.8 (*n* = 9)	5 ± 1.7 (*n* = 7)

Data are presented as mean ± SD. WBCs: white blood cells; QT: quick test; aPTT: activated partial thromboplastin time; AST: aspartate aminotransferase; ALT: alanine aminotransferase; GGT: gamma-glutamyl transferase; ALP: alkaline phosphatase; CRP: C-reactive protein; eGFR: estimated glomerular filtration rate; CPK: creatine phosphokinase; LDH: lactate dehydrogenase. AST, ALT, GGT, ALP, CPK, and LDH values are expressed as multiples of upper limit of normal range. N: number of patients for whom data were available. In cases where nothing is specified, results are provided for entire cohort or subgroup.

**Table 2 jof-11-00462-t002:** Microbial results.

Microbial Results	Total n/N (%)	Specimen-Level Positivity (n/N, %)	Multiple Positive Samples in a Single Patient
Direct microscopy	31/50 (62%)	- BAL: 19/25 (76%) - Bone marrow: 11/22 (50%) - Skin tissue: 1/3 (33%)	- 2 distinct positive samples: 4/28 (14%)
Culture	75/111 (68%)	- BAL: 28/35 (80%) - Bone marrow: 23/30 (77%) - Blood cultures: 18/28 (64%) - Lumbar puncture: 0/9 (0%) - Skin tissue: 2/3 (67%) - Peritoneal liquid: 1/1 (100%) - Colonic biopsy: 1/1 (100%) - Pancreatic tissue: 0/1 (0%) - Lymph node tissue: 2/3 (67%)	- 2 distinct positive samples: 6/37 (16%) - 3 distinct positive samples: 6/37 (16%) - 4 distinct positive samples: 2/37 (5%)
PCR	14/24 (58%)	- BAL: 6/11 (55%) - Bone marrow: 3/4 (75%) - Blood: 3/5 (60%) - Lumbar puncture: 0/3 (0%) - Colonic biopsy: 1/1 (100%) - Lymph node tissue: 1/1 (100%)	- 2 distinct positive samples: 2/19 (11%)
Serum GM	6/25 (24%)	-	-
*Histoplasma* serology	23/29 (79%)	-	-
Contamination with other molds observed in direct examination or culture results	17/92 (18%)	- BAL: 12/35 (34%), including 2 that were completely non-interpretable - Bone marrow: 2/30 (7%), all completely non-interpretable - Blood cultures: 3/28 (11%), including 2 that were completely non-interpretable	-

Total samples refer to the combined total of the BAL, bone marrow, and blood culture samples. Multiple positives refer to the number of patients with more than one positive sample. BAL: bronchoalveolar lavage; PCR: polymerase chain reaction; GM: galactomannan.

**Table 3 jof-11-00462-t003:** Key findings for the three immunosuppression subgroups.

	HIV-Positive (*n* = 20)	HIV-Negative, with Other Cause of Immunosuppression (*n* = 12)	HIV-Negative, Presumed Immunocompetent (*n* = 10)
Age (years)	49 ± 10	55 ± 14.5	54 ± 22
Sex (M/F)	16 (80%)/4 (20%)	7 (58%)/5 (42%)	7 (70%)/3 (30%)
Social vulnerability	16 (80%)	1 (8%)	1 (10%)
Severity criteria	18 (90%)	9 (75%)	4 (40%)
ICU admission	18 (90%)	11 (92%)	4 (40%)
Coinfections (per patient)	2 ± 1.45 (*n* = 18)	0.3 ± 0.5 (*n* = 10)	0.2 ± 0.44 (*n* = 9)
Positive BAL	14 (70%)	9 (75%)	9 (90%)
Positive bone marrow cultures	16 (80%)	9 (75%)	3 (30%)
Positive blood cultures	15 (75%)	6 (50%)	2 (20%)
Positive direct examination	13 (65%)	11 (92%)	4 (40%)
Positive cultures (any site)	18 (90%)	11 (92%)	8 (80%)
Distinct positive culture samples	1.5 ± 0.9	1.75 ± 1.35	1 ± 0.8
Liposomal amphotericin B as 1st-line therapy	18 (90%)	12 (100%)	9 (90%)
Concurrent steroid use	12 (60%)	10 (83%)	9 (90%)
Mortality	8 (40%)	2 (17%)	2 (20%)

The data are presented as the mean ± the SD or the count (%). BAL: bronchoalveolar lavage. Positive BAL, myelogram, and blood cultures refer to the total number of samples that tested positive, either through direct examination or culture. The positive results from direct examination and cultures indicate the number of samples that were positive for each respective method. The number of distinct positive samples in culture reflects the count of unique samples that tested positive in culture.

## Data Availability

The original contributions presented in this study are included in the article. Further inquiries can be directed to the corresponding author.

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
