# Peer review of "Histoplasmosis in Immunocompromised and Immunocompetent Patients in Guadeloupe"

_jof, 2025, doi:10.3390/jof11060462_

Round 1
Reviewer 1 Report
The area of endemicity for Histoplasmosis is not fully understood so this study provides further knowledge to presence in Caribbean. This study relied on proven Histoplasmosis so likely underestimates true incidence which author acknowledge. In addition histoplasmosis diagnosis can be delayed or missed due to poor recognition of clinical manifestations and risk factors highlighting importance of this manuscript. Overall I feel this is a good retrospective cohort study that could be improved by some modifications as noted in comments.
Line 23: “The median diagnostic delay was 3.5 days” – see comments below about Line 199-201 would consider rephrasing to time to diagnosis from admission as to truly say diagnostic delay would also need information about duration of proceeding symptoms and prior health care encounters.
Line 95 – recommend defining Toulouse PCR laboratory database as most people will not be familiar with what this is
Line 96-97 – “No urine samples were submitted to the mycology department for analysis, including Histoplasma PCR 97 testing” - I am not sure how important this statement is as urine testing for Histoplasmosis is only done if you are sending for urine Histoplasma antigen which based on remainder of text is not available so could remove or consider stating that Histoplasma urine antigen testing is not available.
Line 192 – Clinic symptoms, in this section would be helpful to classify disease extent, i.e were they all disseminated histoplasmosis, isolated pulmonary disease, CNS involvement etc
Line 194-195 – “Sixty percent (25/42) required intensive care unit (ICU) admission, although one patient did not meet the severity criteria, and of which 36% (9/42) were primarily admitted to the ICU.” – Consider rephrasing this sentence as it is hard to understand. Are you saying all but one of the individuals admitted to ICU met severity criteria or that 36% of patients met severity criteria and of those with severe disease 60% of patients required ICU admission?.
Line 199-201 “The average diagnostic delay from hospital admission was 3.5 ±10.3 days. All patients, except one, were diagnosed during hospitalization.” – Do you have any data on if patients had prior healthcare encounters? As in were they seen in emergency departments or clinic before this visit or did they have prior hospital encounters for their symptoms? As it is not uncommon for patients to be seen multiple times before the diagnosis of Histoplasmosis is considered as early symptoms can be nonspecific. Regardless of if this information is available would consider rephrasing to the average time to diagnosis from hospital admission rather than average diagnostic delay as this number is likely not capturing if there was a diagnostic delay or not. Another thing that may be helpful to comment on diagnostic delays would be duration of their symptoms prior to admission
Figure 2 – For the head imaging, are you saying these abnormalities are related to histoplasmosis. If so did patients undergo lumbar puncture to evaluate for CNS histoplasmosis? If you do not feel these findings were related to histoplasmosis would consider removing or at least adding comment about how these many be related to other disease processes
Table 2 – please define myelogram culture, are you referring to CSF fluid? If so would group lumbar puncture and myelogram together if these are both CSF. For adenopathy, pancreas and colon are these biopsies or tissue obtained from these areas? If so would change to pancreas tissue, lymph node tissue etc to provide a better representations of where these cultures are from or define them below the table. For the histoplasma serologies do you all have data on positivity of these tests as this would be helpful information. Similar for the serum galactomannan antigen test.
Line 288: Did you see any differences in mortality due to presence of co-infection or not?
Line 321: I am not familiar with this paper but am not sure if it can be extrapolated to Histoplasmosis and wonder if the higher predominance of Histoplasmosis is more likely related to higher predominance of HIV in male population so may consider addition comment about that as well to be more complete.
Line 355-357: “Laboratory findings revealed a systemic inflammatory response and other key abnormalities characterized by cytopenia and elevated levels of liver enzymes, LDH, triglycerides, ferritin, and hyponatremia, indicative of HLH." These are all very nonspecific and do not necessarily mean the patient has HLH, if available would consider adding number of patients who would meet diagnostic criteria for HLH. Otherwise would rephrase and say these can be signs of HLH and were more commonly seen in HIV group and not seen in immunocompetent group to be more clear.
Line 380: In results section you mention Histoplasma serology, is this testing available? If so what were the results.
For limitations – recommend adding comments about underestimating number of cases as the study is limited to histoplasmosis cases with positive cultures, histopath and PCR so likely are missing patients who had probably histoplasmosis given difficulties with diagnosis of histoplasmosis especially in regions where there is not access to Histoplasma testing such as serology and antigen testing.
Please double check to ensure Histoplasma capsulatum and H. capsulatum are properly formatted and italicized.
Reviewer 2 Report
Overall, the article presents relevant information on histoplasmosis in Guadeloupe. Given the limited information available in that country, it is relevant to the scientific community and clinicians in general. Although the overall structure of the article is well-presented, some sections are unclear when read. The authors are encouraged to improve the writing of all sections of the manuscript (introduction, materials, results, etc.) and to significantly improve the presentation of figures and tables, making the manuscript more organized and clear.
The abstract should also be revised and improved for clarity. Make sure to spell the scientific names of the microorganisms correctly, always in italics.
Reviewer 3 Report
The manuscript researches a topic that is very interesting and regarding which little information is available in literature. Therefore, I do think that it should be taken into consideration for publishing. However, there are some major topics that should be addressed beforehand, which I have detailed below.
Major suggestions:
- in the Introduction section, lines 44-51, I can't completely agree with the statements. The authors mostly underline in the introduction the fact that Histoplasmosis is a disease of the immunocompromised patients, when quite the contrary, one of its particularities is the fact that it affects even people with good immunity. Unfortunately, immunocompetent people are completely excluded from the introduction section.
- line 77 - can you please briefly detail the criteria used?
- line 171 - please correct "12/" to "12/33"
- the material and methods section needs to be better structured in my oppinion. The structure doesn't have a logical flow and it just jumps from one subsection to another without having any particular connection between them. For example, the material and methods section starts with inclusion and exclusion criteria, then talks about the ethic's approval, then jumps back to the patients and discusses about data collection. Furtermore, in the data collection section, there is no specific mention of all the data collected, just information added there randomly: the subsection starts by mentioning there was data collected from the institute, but then it randomly says that no urine samples were taken from patients, without any mention of other samples that were collected. Afterwards, in the last paragraph, there is a mention of the patient's HIV status. But we never find out what exact data was taken from the patient file: age? gender? comorbidities? other demographic data? I think the authors should restructure the material and methods section completely as well as all its subsections.
- in the results section, nothing is mentioned about comparing the accuracy of normal cultures vs. molecular diagnosis, yet in the conclusion section, PCR is deemed better than the classical methods. Please correct or add the necessary information to the results section
Minor suggestions:
- can you please write Histoplasma capsulatum in italic? E.g. lines 36, 39, etc. Please also check throught the text
- the English language is sometimes hard to understand. Please revise the text so it sounds more academic. E.g. line 162, the authors wrote "As of November 2024", meaning starting from November 2024. But the study period is from 2014-2024, so it makes no sense discussing the cases after that time slot. Please correct and also check English language all throught the text.
- please add your figures and tables in the text next to the paragraph they are mentioned in
Round 2
Reviewer 3 Report
I would like to thank the authors for taking into consideration my suggestions.
The article has been improved and is suitable for publication in current form.
Author Response
We sincerely thank the reviewer for their positive feedback and for recognizing the improvements made to the manuscript. We are pleased that the revised version meets their expectations and appreciate their support for publication.